# Preparation and Characterisation of Polyurethane Acrylate-Based Titanium Dioxide Pigment for Blue Light-Curable Ink

**DOI:** 10.3390/polym13223977

**Published:** 2021-11-17

**Authors:** Chenglong Wang, Luyang Qiao, Qiang Zhai, Kai Yan, Lili Wang, Jinhuan Zheng

**Affiliations:** 1Engineering Research Center for Eco-Dyeing and Finishing of Textiles, Ministry of Education, Zhejiang Sci-Tech University, Hangzhou 310018, China; wcl_charles@126.com (C.W.); luyang_qiao@163.com (L.Q.); yk950825@126.com (K.Y.); lilichuyingzhanchi@163.com (L.W.); 2China Textile Engineering Society, Beijing 100025, China; zhaiqiang@ctic.org.cn

**Keywords:** polyurethane acrylate-based TiO_2_, polyurethane prepolymer, A-TiO_2_, hyperbranched structures, blue light-curable ink

## Abstract

Herein, a polyurethane acrylate-based TiO_2_ (PU-TiO_2_) was fabricated using a two-step method. First, a polyurethane prepolymer was prepared. Second, PU-TiO_2_ was prepared using amino-modified TiO_2_ (A-TiO_2_). The best synthesis process of the polyurethane prepolymer was when the reaction temperature was 80 °C, the reaction time was 3 h and the *R*-value of the polyurethane acrylate was 2. Next, the influence of the A-TiO_2_ content on the structure and performance of PU-TiO_2_ was examined. The analysis of the rheological properties of the PU-TiO_2_ ink indicated that its viscosity gradually increased as the A-TiO_2_ content increased. The tensile performance of film improved because of the presence of A-TiO_2_. The photo-polymerisation and photo-rheological performance indicated that the PU-TiO_2_ structure changed from a hyperbranched structure with TiO_2_ as the core to a segmented structure, as the A-TiO_2_ content was 3%.

## 1. Introduction

Recently, blue light-curable digital inkjet printing of textiles has been proposed, and this technology has unique characteristics such as energy efficiency, simple processability, broad adaptability, safety and environmental protection [1,2,3]. However, because of the inherent characteristics of pigments, such as high surface energy and low polarity, pigment-based digital inkjet-printing inks have common defects, including the poor dispersion stability of pigment particles in the ink and easy migration and aggregation during the film-forming and film-curing processes. The poor dispersion stability is responsible for a nozzle-clogging problem; furthermore, the pigments, which accumulate on the cured film’s surface, are easily rubbed off. Thus, for pigment-based inkjet printing, improvement in the dispersion stability and colour fastness has become an important aspect for the textile inkjet-printing field.

Based on the abovementioned limitations, encapsulating latex particles on pigment surfaces was considered the most promising method to improve the quality of pigment-based inks [4,5]. Recently, common methods for encapsulating pigment particles, such as layer-by-layer assembly [6,7], mini-emulsion polymerisation [8], emulsion polymerisation [9] and sol-gel method [10,11,12], have been reported. Nevertheless, the dispersion stability of the pigment ink could be improved to some extent with the pigment prepared by encapsulating pigment particles; however, the film still lacks firm forces. In our previous studies [13,14,15], a reactive pigment was proposed, and the double bonds, which could participate in the co-polymerisation of oligomers and monomers to realise the firm anchorage of pigment particles in the polymer cross-linked network structure via covalent bonds, were induced on the pigment surface. Moreover, the colour fastness to the crocking of the printed fabric improved. However, the dispersion stability of the reactive pigment ink did not considerably improve.

Polymer dye [16,17], in addition to the colour body and the polymer skeleton via a covalent bond, had characteristics of the coloured body and film-forming properties of the polymeric material. The polymer dye can effectively improve the migration resistance of polymer dye. Inspired by polymer dyes, a new idea about blue light-curable integrated pigment was innovatively proposed as per the processing properties of blue light-curable pigment-based inkjet printing. In our previous study [18], polyurethane acrylate-based titanium dioxide (PU-TiO_2_) was used to prepare blue light-curable inks as oligomers and white pigments to replace the original white pigment. Furthermore, the influence of TiO_2_ modification on the properties of PU-TiO_2_ was examined; the results demonstrate that, for blue light-curable ink, PU-TiO_2_ was suitable.

Herein, PU-TiO_2_ ink was prepared and its properties were examined. To prepare a polyurethane prepolymer, polypropylene glycol (PPG-1000), isophorone diisocyanate (IPDI) and hydroxyethyl acrylate (HEA) were used. The synthesis process of polyurethane prepolymer was determined using real-time infra-red monitoring and the di-n-butylamine titration method. Next, to prepare PU-TiO_2_, an amino-modified TiO_2_ (A-TiO_2_) was added. PU-TiO_2_ with different amounts of A-TiO_2_ was used to prepare the PU-TiO_2_ ink. Subsequently, the effect of the A-TiO_2_ amount on the properties of the PU-TiO_2_ ink was examined. Furthermore, we examined the rheological and photo-polymerisation properties of the PU-TiO_2_ ink. Moreover, we examined the optical rheological properties of the ink and the breaking strength of the cured film.

## 2. Materials and Methods

### 2.1. Materials

Ditin butyl dilaurate (DBTDL), 2-hydroxyethyl acrylate (HEA), 3-aminopropyltriethoxy silane (APS), and bromocresol green indicator (BCG) were obtained from Aladdin Industrial Corporation, Shanghai, China. Absolute ethanol was purchased from Hangzhou Gaojing Fine Chemical Co. Ltd., Hangzhou, China. Camphorquinone (CQ) and ethyl-4-dimethylaminobenzoate (EDMAB) were obtained from Sigma Aldrich, Shanghai, China. TiO_2_, Isophorone diisocyanate (IPDI) and polypropylene glycol (PPG) were purchased from Aladdin Industrial Corporation, Shanghai, China. Deionised water (>18 MΩ·cm, Millipore Milli-Q, Suzhou, China) was used for all experiments. All chemical reagents were of analytical grade, except BGC, which was of indicator grade, and were used without additional purification.

### 2.2. Methods

#### 2.2.1. Preparation of the PU-TiO_2_ Pigment

All raw materials were dehydrated to ensure that the moisture content was <0.1%. The PU-TiO_2_ pigment was fabricated using a two-step method. First, a certain amount of PPG and IPDI was added to a three-necked flask. The starting temperature was controlled at 10 °C, and the initial -NCO content was determined. Next, the temperature was gradually increased to a specific temperature; furthermore, the DBTDL catalyst was added to the reaction. The -NCO content in the prepolymer system was measured every 30 min. Second, when the -NCO content was reduced to the theoretical value, the reaction temperature was reduced to 50 °C, and a certain number of A-TiO_2_ particles was added to polyurethane prepolymers and reacted at 50 °C for 3 h. Finally, when the -NCO content was less than the theoretical value (when the n_OH_/n_NCO_ of the whole system was 1), the active capping agent, i.e., HEA, was added. When the -NCO content was reduced to 0.5%, the reaction was considered over, and a PU-TiO_2_ pigment was obtained. The detailed process is shown in Figure 1.

#### 2.2.2. Preparation of Blue Light-Curable PU-TiO_2_ Ink

The blue light-curable PU-TiO_2_ ink was composed of a 1 wt% CQ/EDB combination as photo-initiators, and PU-TiO_2_/HEA at a ratio of 5:5 was combined as the ink system. The mixture was dispersed using a covered ultrasonic oscillator (250-W, 40-kHz, Kunshan Ultrasound Co. Ltd, Kunshan, China).

#### 2.2.3. Preparation of Blue Light-Curable PU-TiO_2_ Composite Films

The blue light-curable PU-TiO_2_ ink was evenly coated on a glass sheet. To avoid bubbles, a different glass sheet was slowly covered. Next, sheets underwent a blue light radiation at 16.9 mW/cm^2^ for 8 min. Then, the cured film was peeled from the glass sheet; moreover, the film thickness was appropriately controlled.

### 2.3. Measurements

#### 2.3.1. Fourier Transform Infrared (FT-IR) Analysis

The FTIR spectra of samples were recorded using a Nicolet 5700 FT-IR spectrometer (Thermo Fisher Scientific, MA, USA) at a resolution of 4 cm^−1^ in the range of 4000–600 cm^−1^.

#### 2.3.2. Determination of NCO Content in Synthesis System

The NCO content in the synthesis system was measured using the di-N-butylamine titration method (ISO14896-2009: plastics—polyurethane raw materials—determination of isocyanate content). A reactant amount of 1.0 g was dissolved in an Erlenmeyer flask with anhydrous toluene. Next, 10 mL of di-n-butylamine-toluene solution (0.1-mol/L) was added to the Erlenmeyer flask, shaken and reacted for 20 min. Subsequently, 50 ml of isopropanol and three drops of BCG indicator were added and then titrated with a standard solution of HCl (1 mol/L).

The titration was stopped when the colour changed from blue to light yellow, and the volume of HCl consumed was recorded. The content of the -NCO group was calculated from Equation (1) as follows
(1)$$\mathrm{NCO}\%=\left(V_{0}-V_{1}\right)\times C_{HCl}\times 4.2÷m\times 100$$
where *V*_0_ is the HCl volume before titration (ml), *V*_1_ is the HCl volume after titration (mL), C_HCl_ is the HCl standard solution (mol/L) concentration and m is the mass of the sample (g).

#### 2.3.3. Rheological Performance Analysis

The flow curves of PU-TiO_2_ ink were determined using an MCR 52 senior rotational rheometer(Anton Paar, Graz, Austria). Static test conditions were a temperature of 25 °C and a shear rate of 1–1000 s^−1^, whereas dynamic test conditions were a temperature of 25–70 °C and a shear rate of 200 s^−1^.

#### 2.3.4. Photo-Polymerisation of PU-TiO_2_ Ink

To evaluate the photo-polymerisation of the PU-TiO_2_ ink, photo-differential scanning calorimetry (Q2000 Photo-DSC, American TA Company, New Castle, DE, USA) was used. The heat flow changes in the sample were obtained from Photo-DSC at 25 °C.

#### 2.3.5. Tensile Properties of Blue Light-Curable PU-TiO_2_ Composite Films

The breaking strength and elongation of blue light-curable PU-TiO_2_ composite films were measured using an Instron 3367 material testing machine (Instron, Boston, MA, USA) as per the ISO 1184-1983 standard. The test conditions were a reference temperature of 20 °C, clamp spacing of 20 mm and stretching rate of 10 mm/min.

#### 2.3.6. Photo-Rheological Performance of Blue Light-Curable PU-TiO_2_ Ink

The photo-rheological performance of blue light-curable PU-TiO_2_ ink was evaluated using an MCR 52 rotary rheometer (Anton Paar, Graz, Austria) equipped with an OmniCure S2000 photo-calorimetric accessory (EXFO, Quebec City, QC, Canada). Quartz light guides delivered blue light irradiation from a 200 W Hg arc lamp in the test cell with a 400–500 nm-sized band-pass filter and a 10% attenuation filter. In particular, to determine the linear viscoelasticity regime for the rheological measurement, an oscillatory shear was applied to a pair of 50 mm-sized diameter parallel plates with 10% strain. The gap between these two parallel plates was set to 0.10 mm, and the input angular frequency was set to 62.83 rad/s. Before testing, the samples underwent 30 s equilibration of oscillatory shear at 25 °C without exposure to light. The gel point time (t_gel_), defined as the time at which the storage modulus (G′) is equal to the loss modulus (G″), was determined using a typical dynamic modulus curve.

#### 2.3.7. Dispersion Stability Analysis

Both unmodified and modified TiO_2_ ink materials were diluted in HEA monomer (PU-TiO_2_:HEA = 2:8) and ultrasonically dispersed for 30 min to observe the dispersion stability of the ink.

## 3. Results and Discussion

### 3.1. Effect of Reaction Temperature and Reaction Time on -NCO Reaction

PU-TiO_2_ pigments were synthesised using the prepolymer method; the reaction temperature considerably affected the synthesis efficiency of the polyurethane prepolymer and the system’s viscosity. A lower reaction temperature resulted in a lower reaction rate. However, with an increase in the reaction temperature, the possibility that additional -NCO in the system would participate in side reactions with carbamate or carbamido increased (Figure 2).

With progress in the reaction, the reaction speed became extremely fast as the reaction temperature became extremely high. In addition to easy implosion, -NCO would have more complicated side reactions. Thus, the effect of reaction temperature and reaction time on -NCO reaction was examined; the result is shown in Figure 3.

Figure 3 shows that the reaction rate increased with the increase in reaction temperature. The collision probability of -OH and -NCO also increased, which increased the reaction rate. However, when the temperature reached 90 °C, the side reaction of chain extension occurred in the polymerisation system because of the extremely high temperature, as well as an implosion, resulting in a sharp increase in the system’s viscosity [19]. Therefore, 80 °C was used as the reaction temperature.

The effect of reaction time on the addition reaction between PPG 1000 and IPDI was monitored by FT-IR spectroscopic analysis; the reaction system was tested every 30 min, and the result is shown in Figure 4. It shows that when the reaction time increased, the polyurethane prepolymer exhibited N-H stretching vibration absorption peaks at 1710.8 and 3325 cm^−1^, and the peak at 2262.4 cm^−1^ gradually decreased, indicating that -NCO was continuously consumed till the completion of the reaction.

### 3.2. R-Value Effect on Properties of PU-TiO_2_ Ink

The viscosity of blue light-curable integrated ink was probably one of the most important indexes. Moreover, oligomer properties directly affected the viscosity, photo-polymerisation behaviour and mechanical properties of the cured film. The rheological properties of PU-TiO_2_ ink with different ratios (*R* values) of the soft to hard segments of the polyurethane acrylate are shown in Figure 5 and Table 1. 

Figure 5 shows the temperature effect on the rheological properties of PU-TiO_2_ with different *R* values, and Table 1 shows the *R*-value effect on the viscosity of the PU-TiO_2_ ink. Table 1 shows that the viscosity of the PU-TiO_2_ ink decreased as the *R*-value content increased because the increase in the content of -NCO groups in the system increased the HEA concentration in the capping reaction. Thus, the possibility of the PU-TiO_2_ chain entanglement reduced, thereby reducing the intermolecular force and improving the flow properties [20]. Figure 5 shows that the viscosity of the PU-TiO_2_ ink decreased with an increase in temperature, primarily because the viscosity and temperature behaviours of the PU-TiO_2_ ink conform to the Arrhenius equation. With an increase in temperature, the free volume of polyurethane increased and the degree of chain entanglement decreased, thereby reducing the intermolecular friction and improving rheological properties.

Furthermore, the *R*-value effect on the photo-polymerisation of PU-TiO_2_ ink was examined, and the PU-TiO_2_ ink prepared by PU-TiO_2_ (with different *R* values) and HEA (the ratio of PU-TiO_2_ and HEA was 5:5); the result is shown in Figure 6.

Figure 6 shows the *R*-value effect on the photo-polymerisation performance of PU-TiO_2_ ink. The photo-polymerisation efficiency improved as the *R*-value increased because, as the *R*-value increased, the amount of the end-capping agent (HEA) used increased, which increased the double bond content in the system. The photo-polymerisation efficiency conforms to Equation (2)
(2)$$R_{P}=\frac{k_{p}}{k_{t}^{0.5}}\left[M\right]{\left(\Phi I_{0}\left(1-e^{\left(-2.303\varepsilon \left[CQ\right]d\right)}\right)\right)}^{0.5}$$
where *K_p_* is the chain growth constant, *K_t_* is the chain termination constant, *M* is the double bond concentration, *Φ* is the quantum yield of the photo-initiator, *I*_0_ is the incident light intensity, *ε* is the molar absorption coefficient of CQ and *d* is the thickness of the sample.

Equation (2) shows that the photo-initiation efficiency and double bond concentration have a proportional linear relationship. Note that the photo-initiation efficiency increases as the concentration of the double bond increases [21,22].

Figure 7 shows the *R*-value effect on the tensile properties of the PU-TiO_2_ composite film. The tensile properties of the PU-TiO_2_ composite film gradually increased with the increase in the *R*-value. When the *R*-value was 1.9, the -OH in PPG and the -NCO in IPDI could not completely react in the synthesis process of polyurethane prepolymer. Therefore, the end capping was incompletely performed, which decreased the cross-linking density of the PU-TiO_2_ composite film. Thus, the initial Young’s modulus of the cured film was low, and it was soft when felt by hand. When the *R*-value was 2.2, the HEA-IPDI-HEA small-molecule substance could be generated by the reaction between the excess -NCO in IPDI and HEA. This small-molecule substance participated in the light-curing process, which increased the hardness (felt by hand) and the cross-linking density of the PU-TiO_2_ composite films. Therefore, the material ratio was selected as the *R*-value of 2.

### 3.3. FT-IR

The structure of the polyurethane prepolymer and PU-TiO_2_ was characterised using FTIR spectroscopy; the result is shown in Figure 8.

Figure 8 shows the FT-IR spectrum of the PU-TiO_2_ pigment. A stretching vibration absorption peak belonging to Ti-O occurs at 877.7 cm^−1^, and the peak representing -NCO disappears at 2262.4 cm^−1^, indicating that the PU-TiO_2_ film was successfully prepared.

### 3.4. Effect of A-TiO_2_ Content on Properties of PU-TiO_2_ Ink

The TiO_2_ content considerably influenced the rheological properties of the PU-TiO_2_ ink. Thus, the PU-TiO_2_ ink materials with different A-TiO_2_ amounts were prepared, and the effect of the A-TiO_2_ content on the rheological properties of PU-TiO_2_ was examined; the result is shown in Figure 9.

Figure 9 shows the effect of the A-TiO_2_ content on the rheological properties of the PU-TiO_2_ ink. The viscosity of the PU-TiO_2_ ink gradually increased as the A-TiO_2_ content increased. The influence of the A-TiO_2_ content on the rheological properties of the PU-TiO_2_ ink conformed to the Einstein equation (Equation (3)).
(3)$$\eta =(1+2.5\phi )\eta_{0}$$
where η_0_ is the viscosity of the suspension medium and *ϕ* is the mass fraction of TiO_2_.

Equation (3) shows that the viscosity of the PU-TiO_2_ has a linear relationship with the A-TiO_2_ content. The viscosity of the PU-TiO_2_ gradually increased as the A-TiO_2_ content increased.

Then, the effect of the A-TiO_2_ content on the photo-polymerisation performance of the PU-TiO_2_ ink was examined; the result is shown in Figure 10. The photo-polymerisation efficiency gradually increased as the TiO_2_ content increased. Note that the photo-polymerisation efficiency decreased when the A-TiO_2_ content was 3.0%, primarily because the A-TiO_2_ content increased, and the visible light could not reach the bottom of the sample under the weaker light intensity, which produced a filter effect and affected the photo-polymerisation efficiency. At the same time, when the A-TiO_2_ content was 3.0%, the PU-TiO_2_ structure changed to a segmented structure, which led to a decrease in C=C double bond content.

Then, PU-TiO_2_ composite films were prepared, and the effect of the A-TiO_2_ content on the tensile mechanical properties of PU-TiO_2_ composite film was examined (Figure 11).

Figure 11 shows that the prepared film’s elastic deformation and tensile properties gradually increased when the A-TiO_2_ content increased because A-TiO_2_, as a type of active filler, could produce a strong attraction to the polymerisation system. The polyurethane oligomer was connected to the A-TiO_2_ surface. With the formation of strong covalent cross-linking bonds and physical attraction between chains, A-TiO_2_ grafted with polyurethane uniformly disperses the externally applied pressure load in the film. It reduces the possibility of film breaking and enhances the stretchability of the PU-TiO_2_ composite film.

Usually, the process of photo-polymerisation and cross-linking was accompanied by the gel effect, which was the characteristic and important feature of bulk photo-polymerisation. This phenomenon could be characterised by *G*′ and *G*″. Figure 12a shows that the system was initially in a viscoelastic state because both *G*′ and *G*″ were <20 Pa at the beginning. However, as the irradiation time increased, PU-TiO_2_ and monomer HEA grew, polymerised and cross-linked with each other, and the viscosity gradually increased until it reached the gel point, and *G*′ exceeded *G*″ (viscoelastic transition). Then, *G*′ was always >*G*″. After irradiation at 200 s, the two moduli gradually stabilised, indicating that a stable cross-linked network structure of the photo-curable polymer was formed. The strong nonlinear characteristics of *G*′ provide the possibility of describing the growth behaviour of *G*′ via a typical nonlinear model, given in Equation (4) (Figure 12b).
(4)$$\mathrm{lg}G^{\prime }={G^{\prime }}_{\infty }{\left(1-e^{-\alpha t}\right)}^{\beta }$$
where *G**’∞* represents the platform modulus, and *α* and *β* are fitting parameters.

Indeed, the growth behaviour of time-related *G*′ after a power–law relationship offered an efficient tool with which to quantify the real-time evolution of *G*′ and final storage modulus, *G’**∞* [23,24]. Figure 12c shows that as the A-TiO_2_ content increased, *G*′, which reached a stable cross-linked state, increased. When the A-TiO_2_ content reached 2.5%, *G*′ reached 55 kPa. However, with the increase in A-TiO_2_ content to 3.0%, *G*′ decreased. *G*′/*G**′∞* and its derivative *d*(*G’*/*G’**∞*)/*dt* were defined as the cross-linking degree (*Dc*) and efficiency (*Rc*), respectively. Figure 12d shows that when the A-TiO_2_ content was 3.0%, *Dc* significantly decreased. The cross-linking rate *Rc* demonstrated an inverted “V-shaped” pattern similar to the curve of *Rp* and *t* (Figure 12e), where *Rc* experienced a rapid increase, reached the peak (*Rcmax*), and then gradually decreased. Figure 12e shows that as the A-TiO_2_ content increased, *Rc* gradually decreased. *Rc* was primarily affected by the C=C double bond and molecular chain movement. By analysing the optical rheological properties, the integrated coating formed a hyperbranched structure with TiO_2_ as the core and surface graft coating polyurethane. However, with an increase in A-TiO_2_ content, the hyperbranched structure gradually transitioned to a block structure, which reduced the cross-link density and polymerisation efficiency between PU-TiO_2_. This was consistent with the results of the photo-polymerisation performance analysis.

### 3.5. Dispersion Stability of PU-TiO_2_ Ink

Dispersion stability is an important property of pigment-based ink. Figure 13 shows the effect of the A-TiO_2_ content on the dispersion stability of the PU-TiO_2_ ink. After being placed for three weeks, the dispersion stability of the blue light-curable PU-TiO_2_ ink gradually decreased as the TiO_2_ content in the system increased. When the TiO_2_ content was 3.0%, the sedimentation phenomenon was more obvious than at other TiO_2_ contents. This trend occurred because when the TiO_2_ content in the system is 3.0%, the integrated coating structure transitions from a hyperbranched structure with TiO_2_ as the core to a segmented form. Thus, the steric hindrance effect is reduced, the dispersion stability of TiO_2_ is reduced and the sedimentation phenomenon is obvious.

## 4. Conclusions

Herein, PU-TiO_2_ ink was fabricated using a two-step method. First, to prepare a polyurethane prepolymer, the PPG-1000, IPDI and HEA were selected as the primary raw materials. The best synthesis process was when the reaction temperature was 80 °C, the reaction time was 3 h and the *R*-value of the polyurethane acrylate was 2. Second, PU-TiO_2_ was prepared using A-TiO_2_. The effect of the A-TiO_2_ content on the structure and performance of the PU-TiO_2_ film was examined, and the tensile performance of PU-TiO_2_ film was improved because of the presence of A-TiO_2_. The analysis of the rheological properties indicated that the viscosity of the PU-TiO_2_ ink gradually increased as the A-TiO_2_ content increased. The photo-polymerisation performance and photo-rheological performance indicated that the structure of PU-TiO_2_ changed from a hyperbranched structure with TiO_2_ as the core to a segmented structure, as the A-TiO_2_ content was 3%.

## Figures and Tables

**Figure 1 polymers-13-03977-f001:**
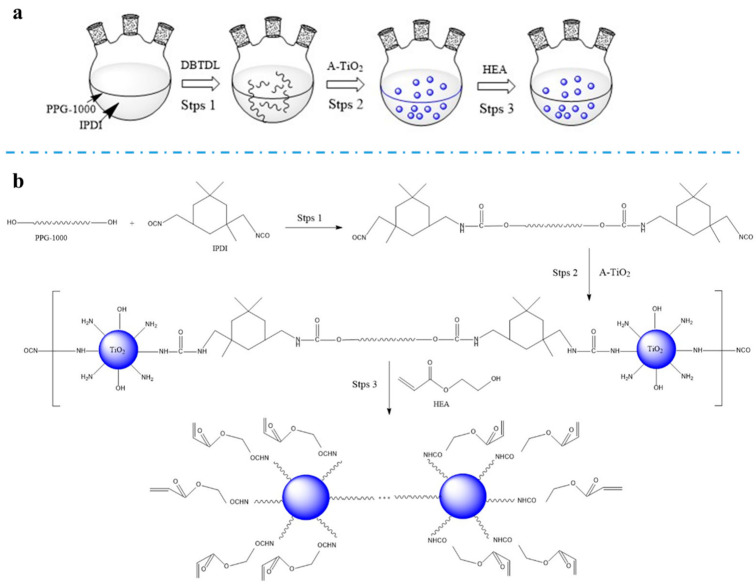
Progress and mechanism for preparation of PU-TiO_2_: (**a**) progress of preparation, (**b**) mechanism of preparation.

**Figure 2 polymers-13-03977-f002:**
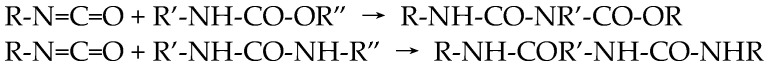
Side reactions with carbamate or carbamido.

**Figure 3 polymers-13-03977-f003:**
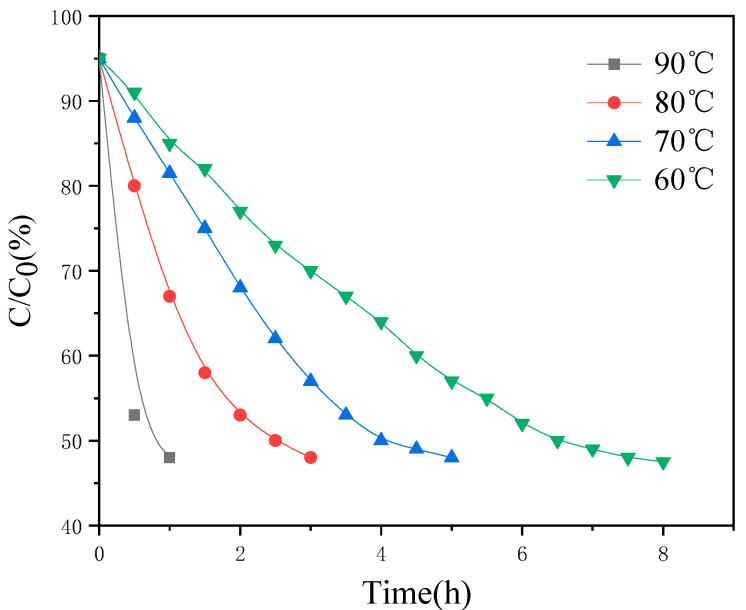
Relationship between -NCO conversion with time and temperature in pre-polymerisation.

**Figure 4 polymers-13-03977-f004:**
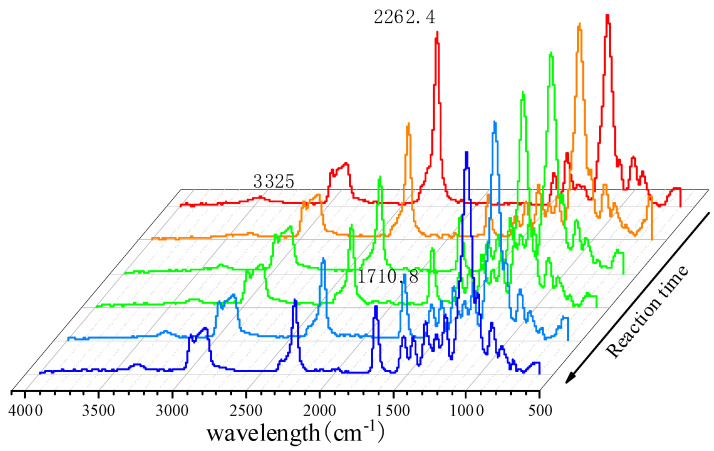
FT-IR spectra of real-time detection of polyurethane prepolymer.

**Figure 5 polymers-13-03977-f005:**
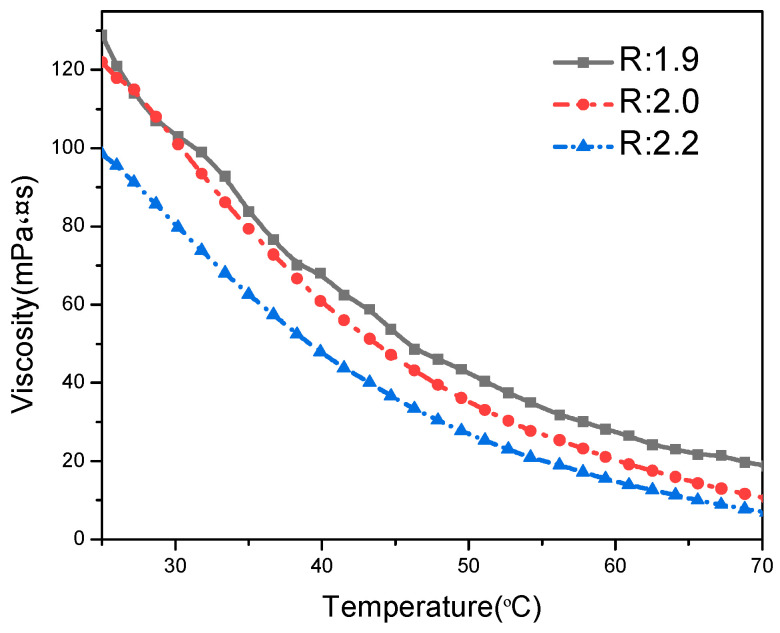
Temperature effect on rheological properties of PU-TiO_2_ ink with different *R*-values.

**Figure 6 polymers-13-03977-f006:**
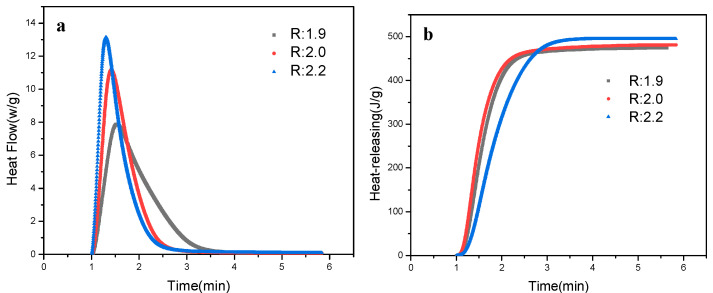
R-value effect on photo-polymerisation performance of PU-TiO_2_ ink: (**a**) photo-polymerisation rates, (**b**) heat releasing.

**Figure 7 polymers-13-03977-f007:**
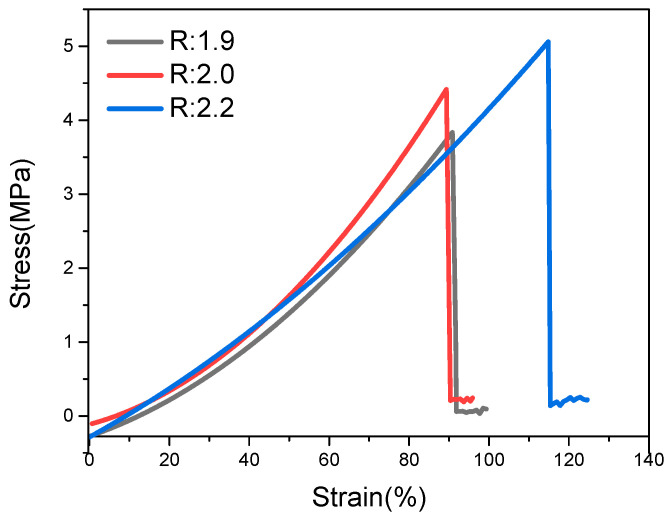
*R*-value effect on tensile properties of PU-TiO_2_ composite film.

**Figure 8 polymers-13-03977-f008:**
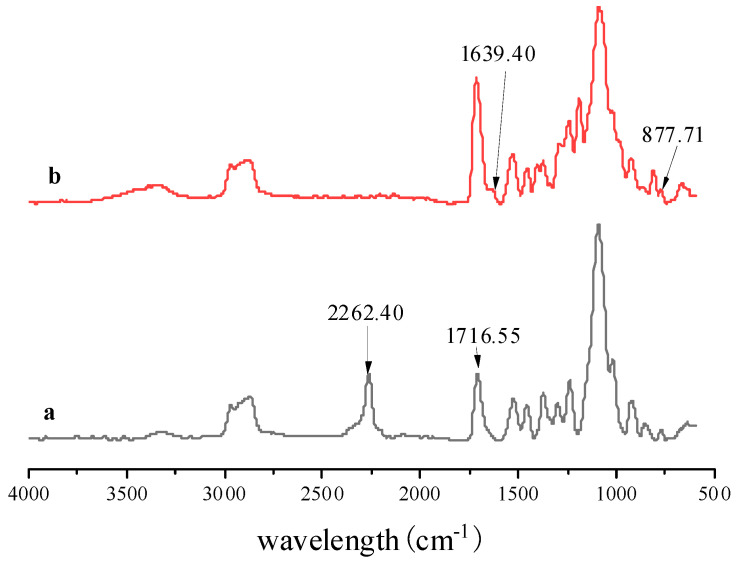
FT-IR spectra of (**a**) polyurethane prepolymer and (**b**) PU-TiO_2_.

**Figure 9 polymers-13-03977-f009:**
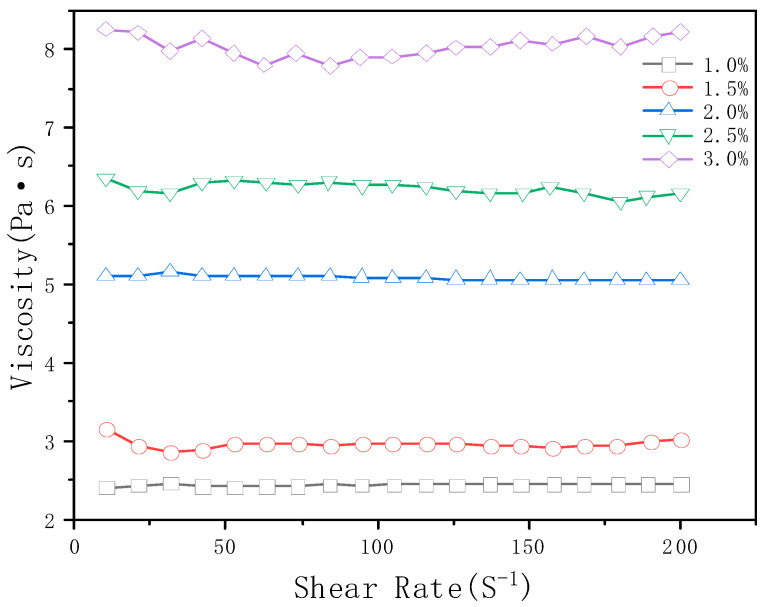
Effect of A-TiO_2_ content on rheological properties of PU-TiO_2_.

**Figure 10 polymers-13-03977-f010:**
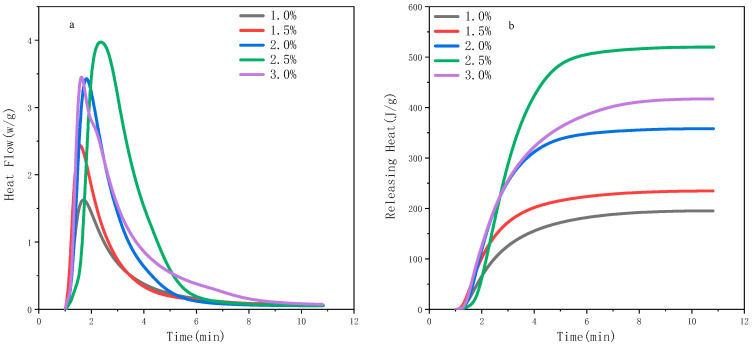
Effect of A-TiO_2_ content on photo-polymerisation performance of PU-TiO_2_ ink: (**a**) photo-polymerisation rates, (**b**) heat releasing.

**Figure 11 polymers-13-03977-f011:**
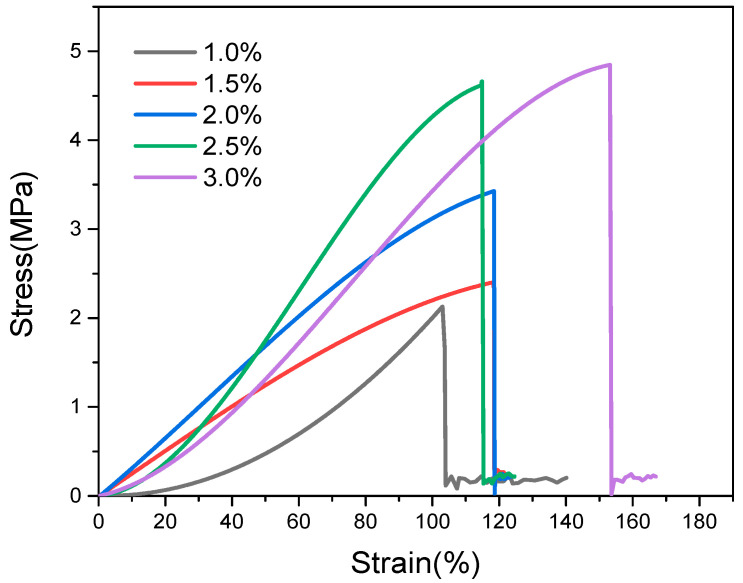
Effect of A-TiO_2_ content on the tensile mechanical properties of PU-TiO_2_ composite film.

**Figure 12 polymers-13-03977-f012:**
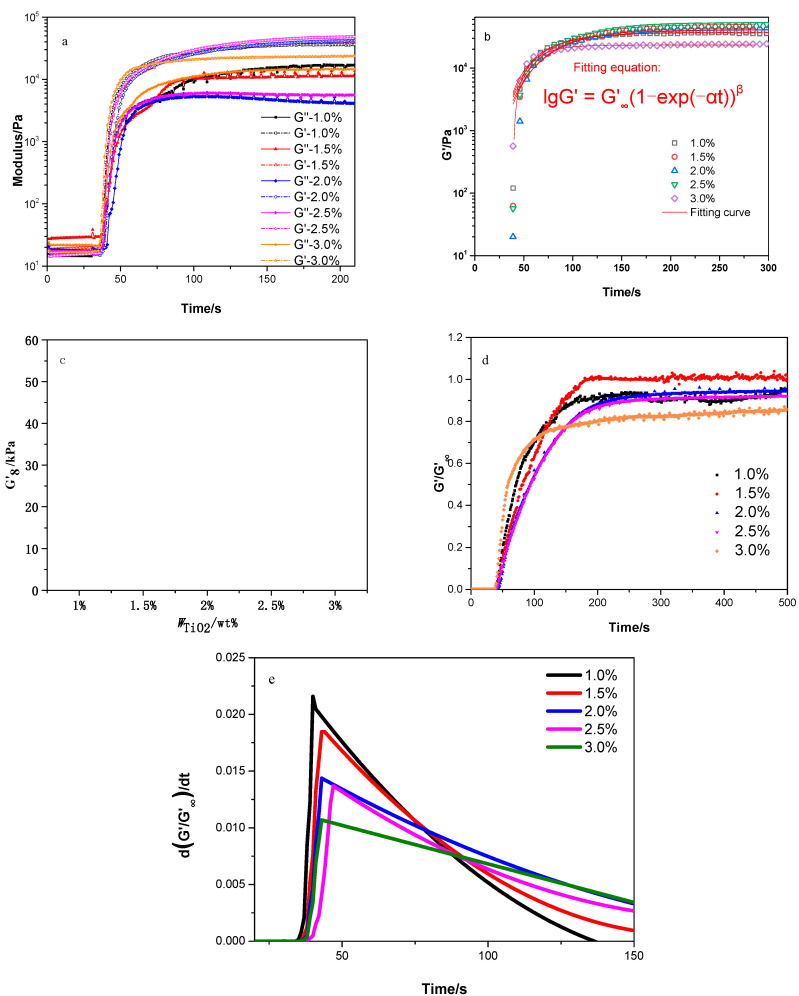
Blue light-induced photo-rheological characteristics of PU-TiO_2_ ink. (**a**) Evolution of *G*′ and *G*″, (**b**) photo-rheological kinetics, (**c**) *G’**_∞_*, (**d**) cross-linking degree and (**e**) cross-linking rate.

**Figure 13 polymers-13-03977-f013:**
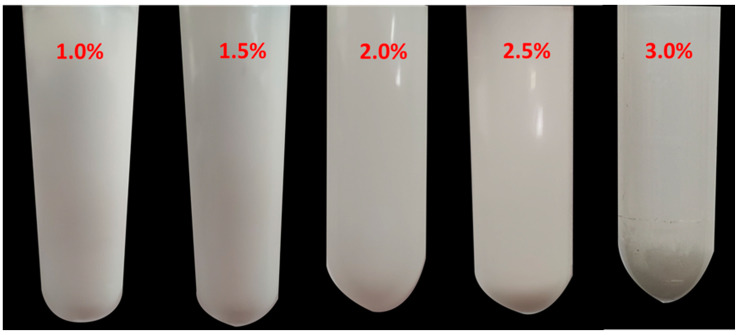
Effect of A-TiO_2_ content on dispersion stability of PU-TiO_2_ ink.

**Table 1 polymers-13-03977-t001:** *R*-value influence on the viscosity of PU-TiO_2_ ink.

*R*-Value	Viscosity (mPa·s)
1.9	120
2.0	108
2.2	98

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
