# Peer review of "Preparation and Characterisation of Polyurethane Acrylate-Based Titanium Dioxide Pigment for Blue Light-Curable Ink"

_polymers, 2021, doi:10.3390/polym13223977_

Round 1

Reviewer 1 Report

the paper has been improved, and it can be published.

Reviewer 2 Report

The paper reports on the fabrication and characterization of a hybrid polymer-titania photosensitive film. The paper could be of interest but it needs major revisions before being considered for publication. The English style and form must be revised by a native language person, since some sentences are not clear at all. The abstract must report only the main quantitative results obtained and not only generic sentences on the work done. The figure captions must described in detail what is represented in the plot or in the scheme shown. For example, in Figure 1 the caption must briefly describe in panel a) and panel b) all the synthesis steps reported. All the steps must be described in detail in the text. The same holds for the other figures. The different colors spectra in Fig. 4 are related to what? Please, add a legend explains the different colors meaning. Fig. 3 and Fig. 4 could be grouped together. At the end of the 3.1 paragraph there is a sentence which seems a recommendation of another referee. Please, cancel these lines. In the captions of Fig. 6 and 10 the heart release is reported but it is clear that is heat release. Please, check and correct these misspelling errors. Which are the values for the fitting parameters alfa and beta in Fig. 12 b) and which are their physical meaning? Please, reports these values and better comment this point. Which is the evidence of hyper branching of the PU-ATiO2 composite at 3% of titania content? Please, report an experimental evidence and related literature data to support this explanation. Conclusions must be improved. Which is the better condition of fabrication? And the better content of titania? Please, quantify the results obtained and avoid generic sentences. For example, which is the increase of viscosity with the A-TiO2 content? And does it fit the requirements of an ink jet nozzle? Without these improvements, the paper cannot be considered for publications.

Round 2

Reviewer 1 Report

  1. The purity of the used chemical should be given in the Section 2.1.
  2. TiO2 should be characterized by size distribution as well as by X-ray diffraction to show the purity of the sample.
  3. Could you give the value of a specific temperature?
  4. How was the dispersion stability monitored? By turbidity measurements?
  5. Line 178. Please remove “Figure 3  shows  the reaction  temperature  effect  on  the  -NCO  ”. In my opinion, this is a repetition.
  6. Line 182-183. “(...) resulting in the system’s viscosity sharply increasing.” Is it your observation? There is no data available as a proof. Please add some values or references.
  7. Line 192-194. Please remove the text.
  8. Figure 3 and Table 1. Standard deviation should be given. Were the experiments repeated or done only once?
  9. Please standardize the format of the headers under the figures.
  10. Why didn't the authors perform a simple spectrophotometer test to show the dispersion stability? That would be a more accurate result.

Reviewer 2 Report

The revised manuscript can be accepted for publication.